# FedTH : Tree-based Hierarchical Image Classification in Federated Learning

**Jae Heon Kim**[*]
School of Computer Science and Engineering
Soongsil University
Seoul 06978, Korea
teeraiser@soongsil.ac.kr

**Bong Jun Choi**[†]
School of Computer Science and Engineering
Soongsil University
Seoul 06978, Korea
davidchoi@soongsil.ac.kr

## Abstract

In recent years, privacy threats have risen in a flood of data. Federated learning was introduced to protect the privacy of data in machine learning. However, Internet of Things (IoT) devices accounting for a large portion of data collection still have weak computational and communication power. Moreover, cutting-edged image classification architectures have more extensive and complex models to reach high performance. In this paper, we introduce FedTH, a tree-based hierarchical image classification architecture in federated learning, to handle these problems. FedTH architecture is constructed of a tree structure to help decrease computational and communication costs, to have a flexible prediction procedure, and to have robustness in heterogeneous environments.

## 1 Introduction

In contemporary society, vast amounts of data are generated and gathered. Not only the amount of data but also the diversity of data has grown. We can have many practical applications and performance improvements from them, but at the same time, it has brought new problems. Network traffic is increasing rapidly, and more security threats have emerged. Federated learning [1] has emerged to handle these issues. Moreover, various Internet of Things (IoT) devices have appeared and spread. They are placed in vast spaces and generate tons of data, including private information. Federated learning is an appropriate state-of-the-art machine learning scheme to secure privacy and reduce communication costs. However, IoT devices usually have low computational and communication resources because of the form-factor feature compared to centralized environments. In addition, they are not optimized for running machine learning. Because of these problems, their limitations must be considered when using machine learning. Meanwhile, cutting-edge architectures have appeared every day in the image classification domain. They are developed with new techniques to enhance image classification performance but are often designed with a larger number of parameters. As a result, for better performance, architectures require greater hardware capabilities.

Because of increasing security threats, form-factor problems, locally busy IoT devices, architecture complexity of image classification, and characteristics of recent data trends, solving the image classification problem on IoT devices becomes more and more difficult. Therefore, we suggest a new scheme FedTH, a tree-based hierarchical image classification architecture and training strategy in federated learning, to solve the mentioned problems and suggest various features.

The main contributions of our approach are summarized as follows:

---

[*]https://sites.google.com/view/davidchoi/home/members
[†]https://sites.google.com/view/davidchoi/home

Workshop on Federated Learning: Recent Advances and New Challenges, in Conjunction with NeurIPS 2022 (FL-NeurIPS'22). This workshop does not have official proceedings and this paper is non-archival.

(1) Compared to traditional approaches having a single large model, FedTH consists of multiple small models. It considers IoT devices' relatively weaker hardware capabilities to reduce computational (time and space complexity) and communication costs on each training round. Furthermore, the space complexity of prediction is also decreased since only one model is used in each prediction step. So, the burden on the client (devices) can be drastically reduced.

(2) In FedTH, the output of all nodes is processable information and clues to prediction. So the classification results can have partial accuracy, whereas existing architectures with black-box features can only classify correctly or incorrectly.

(3) Since the architecture is constructed of a tree structure, it will be robust for non-IID and incremental learning environments. While whole parameters of a single model are affected by non-IID and incremental learning environments, only the shared parents node of biased data or new data will be contaminated in the tree structure, and other nodes will not be affected.

(4) As nodes are independent of other nodes, the server can selectively reinforce weak points of the tree architecture to converge faster and sturdy the architecture. Also, the tree structure is scalable as it can be readily expanded to include additional classes or tailored for the localization of clients.

## 2 Related works

**Image classification in federated learning:** Many research trends of image classification in federated learning of IoT focus on reducing computational and communication costs. [2] proposes algorithms to reduce communication costs in a wireless environment via a fading wireless channel. [3] suggests an effective aggregation method by selecting acceptable models among clients. They concentrate on improving environment elements, but not architecture itself.

**Hierarchical classification:** Hierarchical classification is a natural sequence for humans to classify objects. [4][5] suggest splitting an architecture as sub-models which consist of layers extracting their own features and deriving loss from each sub-model. But, each sub-model is dependent on other sub-models, so their output cannot have a meaning separately. Furthermore, these architectures are chained to a category hierarchy without scalability. Meanwhile, [6] suggests category hierarchy with a visual tree built with taxonomy-independent. From that, the dependency of sub-models' output is cut off. However, even though a taxonomy-independent category hierarchy is more effective for machines, any meaningful information for humans cannot be derived from predictions of sub-models.

## 3 Overview of FedTH

FedTH consists of tree-based hierarchical architecture and a training strategy to train the tree-based architecture on federated learning.

### 3.1 Tree-based hierarchical architecture

FedTH architecture is formed with a tree structure. It is constructed based on a taxonomy-related category hierarchy that follows a traditional human-recognizable classification hierarchy. For example, a *dogs* node is a child of a *mammals* node, and the *mammals* node is a child of an *animals* node. Except for the leaf nodes, every node has its own classification model to classify images to one of its child nodes. Here, the leaf nodes are the classes that are the objective classes to be classified in traditional classification (Fig.1a). The server and clients have the structurally identical architecture (Fig.1b).

### 3.1.1 Prediction procedure

Prediction of the architecture follows a top-down and greedy approach. While the upper node classifies an image of coarse-grained classes with common features, the lower node classifies fine-grained classes with specific features. An image in the prediction procedure basically travel from the root node to a leaf node. Each model of nodes receives the image and predicts it as one of the child nodes of the node. Those steps are repeated until the prediction procedure is finished.

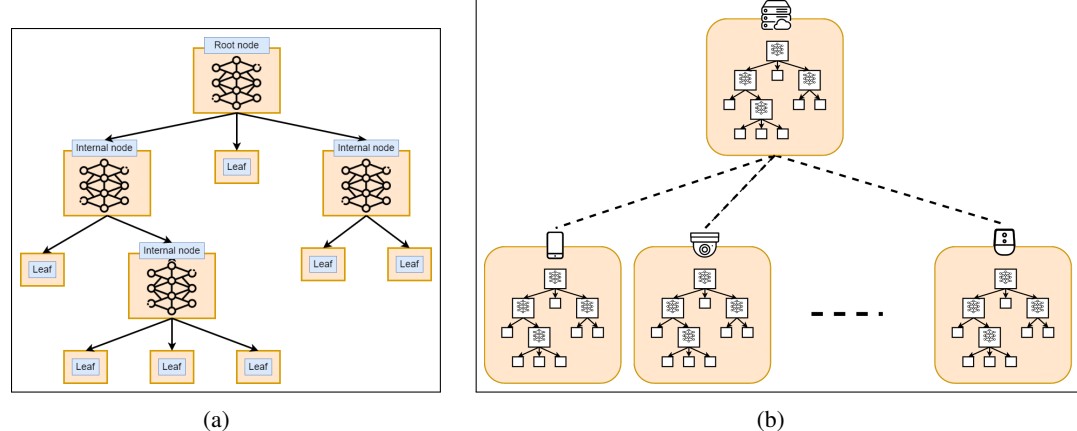

Figure 1: (a) Structure of tree-based hierarchical architecture (b) Entire architectures of server and clients

**Prediction Suspension:** One of the critical problems in hierarchical classification is *error propagation* [6]. If an upper node fails to predict an image correctly at least once, the image loses the chance to classify correctly. To reduce the effect of *error propagation*, *prediction suspension* is suggested. Suppose the probability of prediction on a node is not over a *threshold*. In that case, the node suspends further prediction procedure and finalizes a prediction of the image as a class of the node (Algorithm 1).

---

**Algorithm 1** Prediction on tree-based hierarchical architecture

---

   **Input:** The tree structure, a query image
   **Output:** A predicted class
   $node \leftarrow root$
   **while** $node$ is not $leaf$ **do**
      $P \leftarrow$ (Predicted probabilities of the image with $node$'s model)
      $p \leftarrow max(P)$
      **if** $p \geq threshold$ **then**
         $node \leftarrow$ the $child\ node$ predicted as $p$
      **else**
         **Break**
      **end if**
   **end while**
   **return** A class of the $node$

---

### 3.2 Training procedure on tree architecture in federated learning

During training tree architecture, one of the models is selected and trained on each round. First, the server selects which node will be trained by the scheduler. Next, the server selects which clients will participate in the round. Then, the server distributes a global model of the node, and selected clients receive it. Selected clients train a node model in the client's architecture with the client's local data using a transmitted global model. After finishing the local training step, clients upload the trained model to the server. Finally, the server aggregates clients' models and updates the model of the selected node in tree architecture (Fig. 2). Every node is trained independently.

#### 3.2.1 Tree training scheduler

For faster convergence of the architecture and proper configuration of hyperparameters of each node, we suggest a scheduling algorithm, the **tree training scheduler**. Because of *error propagation*, the impact of the parent node's accuracy affects its children nodes' accuracy down the tree structure. With this characteristic of hierarchical classification, we establish a two-stage strategy in our scheduler.

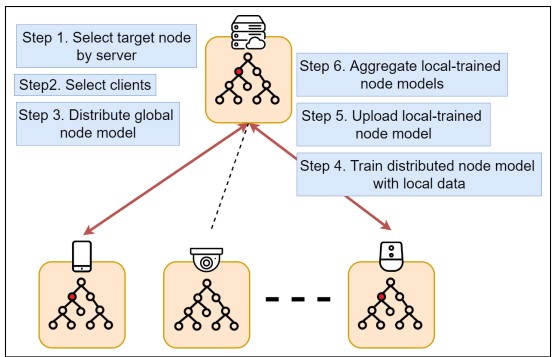

Figure 2: Federated learning process on tree architecture

In the early round of training, the training node is selected with **breadth-first traversal**. From the root node to the leaf node, every node is trained for at least a certain amount of rounds. At this stage, if evaluation loss is not decreased during some rounds, the node will not be selected through this stage. After finishing the stage, the following strategy is priority-based scheduling. The training node is selected by priority, calculated with the node's depth and plateauing of previous training. Upper and active nodes on training are a high priority at this stage. The scheduling method allows quicker convergence with more robust accuracy.

Finally, Algorithm 2 shows how the overall architecture is trained with the server and client. The basic flow of the algorithm follows *FederatedAverage* [1].

---

**Algorithm 2** Federated Learning Procedure with Tree Architecture.

$\beta$ is local minibatch size; $E$ is the number of local epochs, $w_t^{k,i}$ is a weight of $i$-th node in client $k$'s tree architecture on $t$ round; $\mu$ is learning rate

> **Input:** The tree structure
> **Output:** The trained tree structure
> **Server executes:**
> initialize every node's $w_0$
> **for** each round $t = 1, 2, ...$ **do**
>     $node \leftarrow$ Tree-Training-Scheduling()
>     $i \leftarrow index$ of the $node$ on tree
>     $S_t \leftarrow$ Select m clients randomly
>     **for** each client k $\in S_t$ **in parallel do do**
>         $w_{t+1}^{k,i} \leftarrow$ ClientUpdate$(k, w_t^i)$
>     **end for**
>     $w_{t+1}^i \leftarrow \sum_{k=1}^{K} \frac{n_k}{n} w_{t+1}^{k,i}$
> **end for**
>
> **ClientUpdate**$(k, w)$:
> **for** each local epoch $i$ from 1 to $E$ **do**
>     **for** batch $b \in \beta$ **do**
>         $w \leftarrow w - \mu \nabla \ell(w; b)$
>         return $w$ to server
>     **end for**
> **end for**

---

## 4 Experiments

### 4.1 Environment

We evaluated FedTH on the benchmark dataset CIFAR100 [7]. The CIFAR100 dataset has 100 classes, and each class has 500 training images and 100 test images. Class labels consist of coarse labels

(superclass) and fine labels, and each superclass is grouped with 5 fine labels without redundancy. FedTH is implemented on Flower-based federated learning framework, Tensorflow, and Keras. The experiments were done on Intel(R) Core(TM) i9-10980XE and NVIDIA GeForce RTX 3090 with 24 GB memory.

## 4.2    Evaluation criteria

Since FedTH consists of multiple models with a tree structure, the computational cost of a round is not uniform, depending on which node is selected to train by the scheduler. So, an *unit of training progress* (e.g., epochs, rounds, or time) of FedTH is disparate from traditional architectures and schemes. Moreover, the accuracy of a node is affected by the other nodes' accuracy, and *prediction suspension* also should be considered in accuracy.

So, we suggest a new method of evaluation for fair performance measurement. The accuracy of FedTH is calculated with two methods. While *tree accuracy* is based on correspondence between the path of the predicted class and the path of the answer class, *leaf accuracy* only considers whether the prediction is the same as with the answer class. For example, let us assume that the path of a bear node is [animalia, mammals, carnivora, bear]. If a bear image is predicted as carnivora, *tree accuracy* is 0.75 accuracy since carnivora node has a corresponding path to the bear node on [animalia, mammals, carnivora] but not the bear node. However, in this case, *leaf accuracy* is 0 since the image is not predicted as a bear.

A unit of training step is calculated as computational cost. To calculate it, we will use two types of information; the number of images used for training (*IuT*) and the number of multiply-adds operations (*MAdds*). *IuT* represents how many images are used for the training without considering redundancy. For example, if 10 images were trained with a CNN model and 5 epochs, *IuT* is 10×5=50. From the information, we will make two criteria, *IuT* and *IuT*×*MAdds*, to compare computational cost.

## 4.3    Architecture construction

We build a tree architecture with a heuristic category hierarchy for human-recognizable classification. The number of internal and leaf nodes of the tree structure is 20 and 100, respectively, and the maximum and minimum depth of the tree is 4 and 1, respectively. The base node model MobileNetV2 [8] has low computational complexity and few parameters. The initial learning rate is 4e-4, which will be reduced by 75% when evaluation loss does not decrease until 10 steps of node training. We select baseline models with the most similar parameters in terms of the maximum depth of the tree. DenseNet169 [9] has 5 times the number of parameters compared with MobileNetV2. We used provided a model on the Keras application pre-trained on ImageNet. We set the number of clients to 10, of which 5 were selected in the training procedure, and all 10 were selected in the evaluation process. The clients trained with 10 epochs, and the data characteristic of clients is IID.

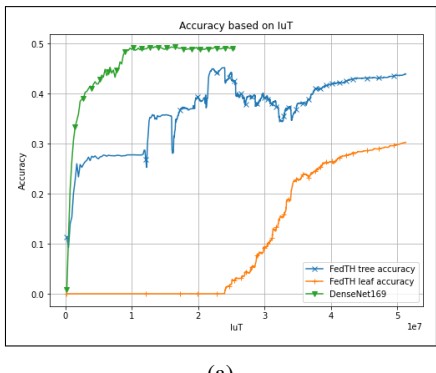
(a)

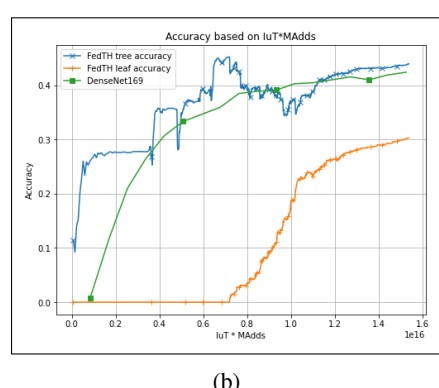
(b)

Figure 3: Test accuracy of FedTH and DenseNet169. (a) Comparison with IuT (b) Comparison with IuT*MAdds (CIFAR1000)

### 4.4 Results

According to the [10], *MAdds* of DenseNet-169 and MobileNetV2 is 3.4 billion and 0.3 billion. Based on only *IuT*, DenseNet-169 converges faster that FedTH with higher accuracy (Fig.3a). However, considering *MAdds*, the convergence speed and accuracy of *tree accuracy* are similar to the tree accuracy of FedTH. In contrast, *leaf accuracy* shows lower performance since it does not count the accuracies achieved in intermediate nodes (Fig.3b). The computational cost of FedTH per round is significantly lower than DenseNet-169. From these results, FedTH has similar convergence speed and accuracy in terms of tree accuracy while reducing the computational cost, reducing communication cost, and providing several new features.

In addition, we expect FedTH to be robust for non-IID and incremental learning environments. The major problem with non-IID is that each client has an uneven data feature. When a model of a client is trained, every client's model has a different bias, so an aggregated model does not converge properly. However, FedTH can retrain the negative effect of non-IID by tree structure which has independence of sub-tree. Different data features of clients train different nodes of tree architecture. Eventually, only parameters affected by non-IID are shared parent nodes, unlike the traditional classification architectures where all parameters are affected. Therefore, we expect that FedTH has strong robustness for non-IID. Furthermore, the characteristic of tree structure also has strengths for an incremental learning environment. Pre-trained models can be disturbed by learning new classes that were not trained before. For the same reason, sub-trees unrelated to additional classes are not affected, so they are preserved from them.

## 5 Challenges

We have proposed a new architecture to provide benefits for resource-constrained devices, flexible procedures, and robustness in heterogeneous environments. However, there remain many challenges to be tackled, as discussed below.

First, the size of the entire architecture, although modularized, is bigger than the previous architecture. Since every node has its model to classify in a tree architecture, the bigger the tree structure is, the bigger the size of the entire architecture is. While the computational and communication costs of one round are reduced, their overall cost is increased. Still, the overall cost could be reduced for each node by taking partial models for specialization.

Second, the effect of *error propagation* can be reduced by *prediction suspension*, but it is not yet completely solved. If the *threshold* for prediction suspension is increased, the effect of *error propagation* will be reduced, but the leaf accuracy will also be decreased. An adaptive *threshold* can achieve better performance. Constructing an efficient category hierarchy is another way to remedy the problem. In our architecture, the upper nodes have relatively higher accuracy but with less meaningful prediction results. The lower nodes have lower accuracy because of *error propagation* but with more meaningful prediction results. A combination taxonomy-independent category for upper nodes and a taxonomy-related category for lower nodes can improve both the reliability and utilization of FedTH.

Last, we have briefly demonstrated the advantages of our approach in a simple setting. To guarantee the robustness of FedTH in diverse environments, more comprehensive experiments with various datasets and base models should be conducted.

## 6 Conclusion

We demonstrated the efficiency and utility of tree-based hierarchical image classification on IoT devices. The tree-based architecture brings unique features from its structure, and these strengths are maximized when used in federated learning. FedTH reduces not only computational and communication costs on each round and space complexity on a prediction but also enables flexible prediction based on accuracy with *prediction suspension*, consolidates robustness for the non-IID and incremental learning environment, and scalability of the architecture. The proposed approach brings new opportunities for image classification in federated learning for IoT devices, along with more challenges to be tackled.

## Acknowledgement

This research was supported by the MSIT, Korea, under the National Research Foundation (NRF), Korea (NRF-2022R1A2C4001270), and the ITRC support program (IITP-2020-2020-0-01602) supervised by the IITP.

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
