# OpenReview forum: "FedTH : Tree-based Hierarchical Image Classification in Federated Learning"
_NeurIPS.cc/2022/Workshop/Federated_Learning — FL-NeurIPS 2022 Poster_

### Official Review · Reviewer_Qx4W · 2022-10-13
**An interesting attempt on hierarchical classification in federated setting**

This paper considers a hierarchical classification, a classification based on a class hierarchy, in a federated setting.
In hierarchical classification, the classifiers are assigned to each node of a tree representing the class hierarchy where the classifier classifies the given input to a certain class or to pass to a descendant node so that the decisions to be made in the later steps.
To realize such a classification in a federated setting, the authors propose to train each classifier in each node step-by-step.
An important step here is the selection of which classifier to train in each of the training rounds.
The authors proposed a two-stage approach for the selection of classifiers.
In the first part, the classifier is chosen in a breadth-first manner in the class hierarchy so that the classifier in shallow nodes to be trained in the early rounds.
In the second part, the classifier is selected by some priority score depending some parameters such as node depth.
The authors reported that the proposed method utilizing the federated setting can yield a model with good performance even if the data are non-IID across the node clients.

### quality, clarity
I found that the paper is not very clear both in its research motivation and technical details.

In the motivation part, the authors claimed that the hierarchical classification is suited for small IoT devices with limited computational resources.
As the authors remarked in "5. Challenges", the use of such a structure actually increases the size of the entire model, which raises a question on the utility of hierarchical classification in small IoT devices.

In the technical part, the authors claimed that the tree training scheduler is the key component in the proposed method.
However, its detail is not explicitly state in the paper.
This will make it difficult for follow up researchers to reproduce the results.

### originality, significance
The attempt of training a hierarchical classification model in a federated setup will be the novelty of the current paper.
However, the significance of the current paper is not very clear in the current state because of its unclear motivation and the lack of technical details.

---

### Official Review · Reviewer_KfV8 · 2022-10-17
**Unclear contribution and performance**

This paper proposed a tree-based hierarchical image classification architecture in federated learning.

The authors claim that, as the main contributions of this paper, the proposed framework can decrease computational and communication costs and have robustness in heterogeneous environments. However, they did not prove it either theoretically or numerically.

The experiments are too few to demonstrate the algorithm performance.

---

### Official Review · Reviewer_SCPe · 2022-10-18
**FedTH : Tree-based Hierarchical Image Classification in Federated Learning**

Strengths:
+ Paper is well written and is easy to follow.
+ The idea of selecting nodes in the tree structure is plausible.
+ The authors present reasonable arguments in favor of their model when it is trained on Non-IDD data
+ The accuracy evaluation results presented in the paper are promising.

Weaknesses:
- The idea of hierarchical model is not very different from a conventional neural network especially from CNN.  In CNN each layer also captures different features of the image.
- Training of hierarchical model is more complex than training a simple neural network because of the complex topology of tree which should be maintained during training. Making changes to the tree is also not simple.
- Each node is a machine learning model and their hyperparamters are shared across all of them, but the presented idea will introduce of new hyperparameter like “number of leaf and internal node” or the maximum length of the tree. These new hyperparameters will be just increasing the complexity of training because you must search for its optimal value.
- The hierarchical structure will become more complex as the number of classes increases because the tree should have a reasonable size to store information about all classes. The argument is based on the assumption presented by the authors in 3.1
- The paper mentions that it will stop at the node where the prediction becomes smaller than the threshold but how one can predict that the output by the next node will always be than the value predicted by parent?
- Only one value is passed to the each of the child when doing prediction. I don’t think passing the single value will be beneficial to the child for deciding.
- The evaluation criteria are presented in term of steps performed by different model which is not a solid criteria because the complexity of each step is not taken account while calculating steps. It also makes fig 3 skeptical because it shows how fast their model converges in term of “steps” not in time.

---

### Decision · Program_Chairs · 2022-10-20

Accept (Poster)